# Development and Implementation of a Quadruple RT-qPCR Method for the Identification of Porcine Reproductive and Respiratory Syndrome Virus Strains

**DOI:** 10.3390/v15091946

**Published:** 2023-09-18

**Authors:** Shengnan Ruan, Wenhui Ren, Bin Yu, Xuexiang Yu, Hao Wu, Wentao Li, Yunbo Jiang, Qigai He

**Affiliations:** 1National Key Laboratory of Agricultural Microbiology, College of Veterinary Medicine, Huazhong Agricultural University, Wuhan 430070, China; ruan@webmail.hzau.edu.cn (S.R.);; 2The Cooperative Innovation Center for Sustainable Pig Production, Huazhong Agricultural University, Wuhan 430070, China

**Keywords:** PRRSV, genotyping, quadruple, abortion, respiratory distress, diagnosis

## Abstract

Background: Porcine reproductive and respiratory syndrome virus (PRRSV) causes porcine reproductive and respiratory syndrome (PRRS), leading to abortion in sows and respiratory distress in breeding pigs. In China, PRRSV1 and PRRSV2 are the two circulating genotypes in swine herds, with distinct virulence. PRRSV2 further consists of classical (C-PRRSV2), highly pathogenic (HP-PRRSV2), and NADC30-Like (N-PRRSV2) subtypes. The diversity of PRRSV poses challenges for control and eradication, necessitating reliable detection assays for differentiating PRRSV genotypes. Methods: A new TaqMan-based RT-qPCR assay with four sets of primers and probes targeting conserved regions of the ORF7 and NSP2 genes of PRRSV was developed, optimized, and evaluated by us. Reaction conditions such as annealing temperature, primer concentration, and probe concentration were optimized for the assay. Specificity, sensitivity, repeatability, stability, limit of detection (LOD), concordance with the reference method were evaluated for the assay. Results: The assay could detect and type PRRSV1, C-PRRSV2, HP-PRRSV2, and N-PRRSV2 simultaneously with 97.33% specificity, 96.00% sensitivity, 12 copies/μL LOD, 97.00% concordance with reference assays. We applied the assay to 321 clinical samples from swine farms in China. The assay successfully detected and typed 230 PRRSV-positive samples, with 24.78% (57/230) of them further confirmed by ORF5 gene sequencing. The prevalence of PRRSV subtypes among the positive samples was as follows: C-PRRSV2 (15.22%), HP-PRRSV2 (23.48%), and N-PRRSV2 (61.30%). Two samples showed coinfection with different PRRSV subtypes. Conclusion: The quadruple RT-qPCR assay is a powerful tool for detecting and typing the currently circulating PRRSV strains in Chinese swine populations. It can assist in the surveillance of PRRSV prevalence and the implementation of prevention and control strategies.

## 1. Introduction

Porcine reproductive and respiratory syndrome (PRRS) is a viral disease that is caused by the porcine reproductive and respiratory syndrome virus (PRRSV), which leads to reproductive disorders in sows and respiratory distress in pigs [1,2]. PRRSV infection causes immunosuppression, increasing the susceptibility to secondary infections, which often results in a higher mortality of infected animals [3]. Additionally, post-infection, PRRSV viremia can last up to two months, which complicates the virus eradication in pig farms.

In 1987, a disease-causing reproductive and respiratory disorder in pigs was initially reported in the United States. This severe outbreak led to a decline in healthy pig populations due to high mortality. In 1991, Wensvoort identified a pathogen RNA virus responsible for the reproductive and respiratory disease [4], which was later termed porcine reproductive and respiratory syndrome (PRRS) by European researchers. The analysis of the genome nucleotide sequences of PRRSV strains isolated from Europe and North America indicated a difference of about 44% among them. Consequently, PRRSV was categorized into type 1 (European genotype) and type 2 (North American genotype) basing on their genetic variation and geographic origin. The Lelystad virus (LV) and ATCC VR-2332 strains represent type 1 and type 2 PRRSV, respectively. PRRSV quickly spread to Europe, Asia, and eventually all other regions [5].

PRRSV is a highly mutable and recombinant single-stranded RNA virus [6,7]. The worldwide spread of PRRSV strains through international pig breeding and pork trade has caused significant economic losses to the pig industry in China [8]. Moreover, the genotypes of PRRSV strains in China have become increasingly diverse due to the spread of multiple strains. Within PRRSV2, nine lineages can be delineated basing on the ORF5 gene sequence. In China, CH-1a-like (8.7), VR2332-like (5.1), QYYZ-like (3.5), NADC30 (1), and European type I isolates are the five identified lineages. Deletions are frequently observed in PRRSVs, including the 30-amino-acid (aa) discontinuous deletion in the NSP2 protein of the highly pathogenic PRRSV strain found in China, the 131-aa discontinuous deletion in the NSP2 protein of NADC30-like PRRSV, and the 100-aa continuous deletion in the NSP2 protein of NADC34-like PRRSV [9,10]. Recent studies have shown that newly isolated PRRSV strains sharing similarities with NADC30-like belong to sublineage 1.8 and are becoming increasingly prevalent. Notably, different PRRSV strains exhibit variations in pathogenicity. Moreover, the cross-protection effectiveness of PRRSV vaccines is limited, which makes the rapid and precise identification of PRRSV genotypes crucial for monitoring the prevalence and genetic evolution of PRRSV, selecting the appropriate vaccine, and finally formulating a proper prevention and control plan for PRRS.

Current diagnostic methods for PRRSV infection include virus isolation, conventional or real-time RT-PCR, and other molecular diagnostic methods [11,12,13]. Although virus isolation is considered the reference standard to confirm the PRRSV infection, the method has limited sensitivity, a lengthy process, and demands technical equipment and expertise, which makes it impractical for rapid diagnosis. On the other hand, serological assays, such as ELISA, are mainly used to detect antibodies against PRRSV after the acute phase has passed or post-vaccination. These assays are not suitable for early diagnosis since PRRSV-specific antibodies are often not detectable up to 7 days post-infection [11,12,13]. In contrast, PCR and qPCR molecular assays are widely popular to detect PRRSV in clinical samples due to their high sensitivity, specificity, and short detection time [14,15,16].

Compared to monoplex qPCR, multiplex qPCR enables the simultaneous detection of multiple PRRSV strains in a more expedient manner. Thus, in pursuit of a more efficient identification method for the four genotypes of PRRSV isolates (PRRSV1, C-PRRSV2 (classical), HP-PRRSV1 (highly pathogenic), and N-PRRSV2 (NADC30-like)), a multiplex RT-qPCR assay based on genomic nucleotide were developed. This assay targets several viral genes simultaneously and therefore provides superior sensitivity and specificity in clinical specimens. This method holds great promise for diagnosing coinfections and monitoring the spread of PRRSV in swine populations.

## 2. Materials and Methods

### 2.1. Viruses and Clinical Samples

This investigation mainly focused on four genotypes of PRRSV strains, particularly PRRSV1 (LV, M96262.2), C-PRRSV2 (CH-1a, AY032626.1), HP-PRRSV2 (TJ, EU860248.1), and N-PRRSV2 (HNGZ2201-NH). To assess the specificity of the RT-qPCR assay, we utilized several species of viral pathogens, such as classical swine fever virus (CSFV, HM175885.1), porcine epidemic diarrhea virus (PEDV, KT021232.1), pseudorabies virus (PRV, AF306511.1), porcine parvovirus (PPV, AY789532.1), porcine circovirus type 2 (PCV2, FJ598045.1), and porcine circovirus type 3 (PCV3, OM032567.1).

In total, 321 clinical samples were collected from pig farms all over China, including 20 cultured viruses; 60 oral, nasal, and anal swabs; 105 tissue specimens; 96 environmental swabs; and 40 serum samples.

### 2.2. Primers and Probes

In total, 150 PRRSV complete genomes, including 50 PRRSV1 and 100 PRRSV2 genomes, were aligned using Mega7 and Megalign software. The PRRSV ORF7 gene demonstrates a highly conserved region among various strains; however, notable differences exist within this region between PRRSV1 and PRRSV2. Consequently, one primer pair (ORF7-F/ORF7-R) and two probes (PRRSV1-ORF7-ROX and PRRSV2-ORF7-FAM) were designed using various software, including Snapgene, Primer Premier, and Oligo7. This design strategy enabled the differentiation between PRRSV1 and PRRSV2. Additionally, two pairs of primers (C-PRRSV2-NSP2-F/C-PRRSV2-NSP2-R and HP-PRRSV2-NSP2-F/HP-PRRSV2-NSP2-R) and two probes (C-PRRSV2-NSP2-CY5 and HP-PRRSV2-NSP2-HEX) were developed to recognize PRRSV subtypes C-PRRSV2, HP-PRRSV2, and N-PRRSV2 based on the relatively conserved region of the NSP2 gene across the three distinct subtypes (Figure 1). The primers and probes were synthesized by synthesized by Accurate Biotechnology(Hunan) Co., Ltd., Changsha, China.

### 2.3. RNA Extraction and Reverse Transcription 

Viral nucleic acids were extracted from cultured viruses; clinical samples oral, nasal, and anal swabs; tissue specimens; environmental swabs; and serum samples—using the DNA/RNA Extraction Kit (Prepackaged) (Nanjing Vazyme Biotech Co., Ltd., Nanjing, China) following the manufacturer’s instructions. The extracted viral nucleic acids were eluted in 100 μL of nuclease-free double-distilled water (ddH_2_O) and stored at −80 °C until further use. The synthesis of cDNA was carried out using HiScript^®^ III All-in-one RT SuperMix Perfect for RT-qPCR (Nanjing Vazyme Biotech Co., Ltd., Nanjing, China), with a reverse transcription program including reaction incubation at 50 °C for 15 min, followed by 85 °C for 5 s.

### 2.4. Establishment of the Standard Curve 

Respective standard plasmid solutions were prepared in varying concentrations (Table 1). The target fragment was amplified using ApexHF HS DNA Polymerase FS Master Mix (dye plus) synthesized by Accurate Biotechnology (Hunan) Co., Ltd., Changsha, China. Specifically, nucleic acid fragments (PRRSV1-ORF7, PRRSV2-ORF7, C-PRRSV2-NSP2, HP-PRRSV2-NSP2, and N-PRRSV2-NSP2) of each subtype strain were amplified, cloned into the PMD-18T vector, and then transformed into competent DH5α cells. The plasmids were purified using a HiSpeed Plasmid Mini Kit (Qiagen), and the DNA concentration was determined according to OD_260nm_ using a Nano_200_ spectrophotometer (Aosheng). The plasmid concentration was subsequently converted into copy number using the formula y (copies/μL) = (6.02 × 10^23^) × plasmid concentration ng/μL × 10^–9^ DNA)/(DNA length × 660).The concentrations of PRRSV1-ORF7, PRRSV2-ORF7, C-PRRSV2-NSP2, HP-PRRVS2-NSP2, and N-PRRSV2-NSP2 plasmids extracted in diethylpyrocarbonate (DEPC) H_2_O were 332.21, 219.50, 190.44, 224.39, and 171.17 ng/µL, respectively. The A_260nm_/A_280 nm_ ratios for these plasmids were 3.02, 2.95, 2.07, 2.87, and 1.95, respectively. The copy numbers for each plasmid were 8.09 × 10^10^, 4.77 × 10^10^, 5.01 × 10^10^, 4.07 × 10^10^, and 4.23 × 10^10^ copies/µL. Each plasmid solution was subjected to 10-fold gradient dilution in DEPC H_2_O and then stored at −80 °C for subsequent use in the study. The standard curve had a linear relationship between the Ct values and the cDNA concentration. Therefore, serial dilutions of a plasmid of a known concentration were used to build a standard curve.

### 2.5. Optimization of Amplification Conditions

To determine the optimal reaction temperature, primers, and probe concentrations, a fixed reaction system and program were utilized to investigate the optimal DNA annealing temperature from 53 to 60 °C. The primers and probes were diluted in DEPC H_2_O to prepare six gradients of 0.2, 0.3, 0.4, 0.5, 0.6, and 0.7 µmol/L. Orthogonal experiments were employed to determine the optimal combination of primers and probes concentration. The RT-qPCR reaction conditions were as follows: 50 °C for 2 min, 95 °C for 30 s, and 40 cycles of 95 °C for 15 s, and finally, 59 °C for 30 s. The optimal amplification condition was determined for the absence of a nonspecific signal, high fluorescence intensity, low Ct value, and nearly 100% amplification efficiency.

### 2.6. Specificity, Sensitivity, Repeatability, and Stability 

To assess the specificity of the RT-qPCR method, RNA was extracted from CSFV, PEDV, and PRRSV strains, including C-PRRSV2, HP-PRRSV2, and N-PRRSV2. Reverse transcription was performed, and the resulting cDNA was used as the reaction template for PCR. In addition, PPV, PCV2, PCV3, PRV, and recombinant PRRSV plasmids (PRRSV1-ORF7, PRRSV2-ORF7, C-PRRSV2-NSP2, HP-PRRSV2-NSP2, and N-PRRSV2-NSP2) were extracted and used as templates to evaluate the specificity of the established method. Each sample was tested in triplicates. The PRRSV (LV) nucleic acid was obtained from a synthetic plasmid Puc19 based on the sequence. The porcine reproductive and respiratory syndrome virus (PRRSV) (CH-1a and TJ), classical swine fever virus (CSFV), and porcine parvovirus (PPV) nucleic acids were sourced from commercial vaccines. Other pathogens were isolated and stored as strains in our laboratory.

To assess the sensitivity of the RT-qPCR method, recombinant plasmids (PRRSV1-ORF7, PRRSV2-ORF7, C-PRRSV2-NSP2, HP-PRRSV2-NSP2, and N-PRRSV2-NSP2) were serially diluted from 10^9^ to 10^1^ copies/µL. Additionally, nucleic acid was extracted from all the PRRSV strains (C-PRRSV2, HP-PRRSV2, and N-PRRSV2) and serially diluted 10-fold to six different concentrations. These were utilized to evaluate the method’s sensitivity. The lowest detectable nucleic acid copy number was defined as the lowest detection limit of the method.

To assess the reproducibility (intrabatch) and stability (interbatch) of the RT-qPCR method, standard plasmids (PRRSV1-ORF7, PRRSV2-ORF7, C-PRRSV2-NSP2, HP-PRRSV2-NSP2, and N-PRRSV2-NSP2) diluted to 10^5^, 10^4^, and 10^3^ copies/μL were detected, and the corresponding detection results (Ct values) were recorded at different time points. The intrabath and interbatch coefficients of variation (CVs) of less than 5% denoted high repeatability and stability, respectively.

### 2.7. The Standard Curve and the LOD of This Multiplex Amplification

To avoid the cross-interference within different gene types of PRRSV in experiments, different subtypes of nucleic acids were mixed for testing after the single nucleic acid test. Four plasmids (PRRSV1-ORF7, PRRSV2-ORF7, C-PRRSV2-NSP2, and HP-PRRSV2-NSP2) with concentrations of 10^8^ copies/μL were mixed and subsequently diluted 10-fold to obtain seven concentrations ranging from 10^8^ to 10^2^ copies/μL at equimolar concentrations. This series of different concentrations of plasmids were amplified to observe the concentration-signal intensity relationship curves of each gene segment. 

To determine the minimum detection limit of this method in detecting mixed plasmids and mixed virus cDNA, the mixture containing four plasmids and three virus cDNAs (C-PRRSV2, HP-PRRSV2, and N-PRRSV2) with known concentrations was separately diluted to concentrations ranging from 10^7^ to 10^0^ copies/μL and 10^5^ to 10^0^ copies/μL at equimolar concentrations. All reactions were performed in triplicate.

### 2.8. Simulation of Clinical Sample

To evaluate the efficiency of this method in clinical samples, four mixed plasmids with a range of concentrations from 10^6^ to 10^0^ copies/μL and three viruses with a range of concentrations from 10^5^ to 10^0^ copies/μL were mixed into 200 μL of serum samples and 0.5 g of minced lung tissue samples separately to simulate the clinical samples. These simulants were extracted, and then they were detected by our method.

### 2.9. Comparison with Referent RT-qPCR

To validate the sensitivity, specificity, and agreement of the developed multiple RT-qPCR method, a total of 100 samples of various types—including viral solutions, swabs, lungs, anticoagulated bloods, and sera—were tested. The multiple RT-qPCR assay was compared with the national reference method (GBT 35912-2018) for detecting PRRSV, a clinical widely used method [18]. Briefly, RNA was extracted from the respective sample and amplified with the developed method using specific primers. The resulting amplicons were purified for ORF5 sequenced using sanger sequencing technology. The obtained sequences were aligned with reference sequences using Megalign and analyzed for genetic variability and phylogenetic relation.

### 2.10. Clinical Application

From January to June 2022, a total of 321 samples suspected of PRRSV infection were examined by the developed multiple RT-qPCR assay. The collected samples from swine farms comprised various specimens including oral, nasal, and anal swabs, lung tissues, and sera. Subsequently, all the positive samples were further confirmed by ORF5 gene sequencing, which is considered the gold standard method for differentiating PRRSV. Clinical samples were collected from the Animal Disease Diagnosis Center at Huazhong Agricultural University.

## 3. Results

### 3.1. Designation of Primers and Probes

The sequences and positions of the primers and probes used RT-qPCR method are summarized in Table 2.

### 3.2. Identification of Recombinant Plasmids

The plasmids constructed in this study were employed to develop the RT-qPCR method and subsequently subjected to conventional PCR detection. The PCR results showed successful amplification of targeted fragments using the specific primers. The expected sizes of the amplified fragments, as determined by PCR, were 398 bp, 433 bp, 1041 bp, 951 bp, and 648 bp for the PRRSV1-ORF7, PRRSV2-ORF7, C-PRRSV2-NSP2, HP-PRRSV2-NSP2, and N-PRRSV2-NSP2 subtypes and genes, respectively (Figure 2). 

### 3.3. Optimization of Reaction Conditions

#### 3.3.1. Annealing Temperature

The RT-qPCR assay in this study was set up for a gradient of annealing temperatures ranging from 53 to 60 °C at a 1 °C difference per reaction condition ramp. The four standard plasmids—PRRSV1-ORF7, PRRSV2-ORF7, C-PRRSV2-NSP2, and HP-PRRSV2-NSP2—were amplified using RT-qPCR at a template concentration of 10^6^ copies/µL. The optimal annealing temperatures for the respective plasmid were determined by analyzing the amplification results obtained at different temperatures (Figure 3a). Specifically, PRRSV1-ORF7, PRRSV2-ORF7, and HP-PRRSV2-NSP2 had an optimal annealing temperature of 57 °C, while C-PRRSV2-NSP2 was best amplified at 59 °C. The optimal annealing temperature was 59 °C for the simultaneous amplification of all four plasmids. The detailed data for Figure 3a,b are shown in Appendix A.

#### 3.3.2. Optimal Primer and Probe Concentrations

Basing on the orthogonal test approach, the optimal primers and probe concentrations obtained from the quadruple system were 0.40 and 0.30 μmol/L, respectively (Figure 3b). The final optimized reaction system and the corresponding RT-qPCR program are listed in Table 3.

### 3.4. Establishment of Standard Curve

The quadruple RT-qPCR exhibited a standard curve slope ranging from −3.46 to −3.15. The correlation coefficient (R^2^) ranged from 0.99 to 1.00, and the amplification efficiency (E) ranged from 94.5% to 107.5%. Each dilution had 3 replicates (Figure 4a–d). A linear correlation between copy number and Ct value was observed. The corresponding linear equations for copy number (x), Ct value (y), and the correlation coefficient R^2^ were as follows: PRRSV1-ORF7: y = −3.461x + 44.794, R^2^ = 0.995, E = 94.5%. PRRSV2-ORF7: y = −3.242x + 38.208, R^2^ = 1.000, E = 103.4%. C-PRRSV2-NSP2: y = −3.155x + 40.311, R^2^ = 1.000, E = 107.5%. HP-PRRSV2-NSP2: y = −3.288x + 44.096, R^2^ = 0.999, E = 101.4%.

### 3.5. Specificity Tests

Nucleic acids were extracted from various viruses and subjected to amplification using the established method. However, only the PRRSV-positive templates and their corresponding plasmids produced fluorescent signals. No positive signals were observed for amplification of other viruses such as CSFV, PEDV, PRV, PPV, PCV2, and PCV3. In the case of C-PRRSV2 (VR2332 strain), HP-PRRSV2 (TJ strain), and N-PRRSV2 (NADC30-like strains) as templates, the positive signals were detected only for the strains that were matched by detection primers and probe sets, indicating good inter- and intraspecificity against the viruses (Figure 5).

### 3.6. Sensitivity Test

For the sensitivity test, 10-fold diluted standard plasmids (ranging from 10^9^ to 10^1^ copies/µL) were utilized. The results indicated that the lowest detectable copy numbers of PRRSV1-ORF7, PRRSV2-ORF7, C-PRRSV2-NSP2, and HP-PRRSV2-NSP2 were 80, 47, 50, and 40 copies/μL, respectively (Figure 6a). The supernatants obtained from C-PRRSV2, HP-PRRSV2, and N-PRRSV2 infected cells were used for extracting the viral nucleic acids, which were diluted 10-fold and followed amplicating by RT-qPCR. The RT-qPCR detection limits of the viral nucleic acids for C-PRRSV2, HP-PRRSV2, and N-PRRSV2 strains were 13, 14, and 12 copies/μL, respectively (Figure 6b). These results indicate the high sensitivity of the test.

### 3.7. Repeatability Test

To evaluate the repeatability and stability of the RT-qPCR method, plasmids (ranging from 10^5^ to 10^3^ copies/µL) were employed. The results showed that the intra- and interbatch coefficients of variation (CV) ranged from 0.10% to 3.90% and 0.09% to 4.69%, respectively, and were less than 5%. These findings indicated that the established RT-qPCR method had good repeatability and stability (Table 4).

### 3.8. Standard Curve and LOD

The curves showed linear relationships within the range of our measurements. The corresponding linear equations for copy number (x), Ct value (y), and the correlation coefficient R^2^ were as follows: PRRSV1-ORF7: y = −3.104x + 41.818, R^2^ = 0.952, E = 110.0%. PRRSV2-ORF7: y = −3.467x + 39.242, R^2^ = 0.994, E = 94.3%. HP-PRRSV2-NSP2: y = −3.348x + 41.553, R^2^ = 0.983, E = 98.9%. C-PRRSV2-NSP2: y = −3.156x + 38.805, R^2^ = 0.974, E = 107.4% (Figure 7a). The results indicated that the detection limits of all the four plasmids (PRRSV1-ORF7, PRRSV2-ORF7, C-PRRSV2-NSP2, and HP-PRRSV2-NSP2) were 1× 10^2^ copies/μL, and the detection limits of all the three virus cDNA (C-PRRSV2, HP-PRRSV2, and N-PRRSV2) were 1 × 10^1^ copies/μL (Figure 7b).

### 3.9. LODs of Clinical Simulants

The LODs of serum samples and lung tissue samples simulated by plasmids were determined to be 1 × 10^2^ copies/μL and 1 × 10^4^ copies/μL. The LODs of serum samples and lung tissue samples simulated by viruses were determined as 1 × 10^2^ copies/μL (ORF7), 1 × 10^3^ copies/μL (C-PRRSV2-NSP2), and 1 × 10^4^ copies/μL (HP-PRRSV2-NSP2) (Figure 8a,b).

### 3.10. Comparison between the National Reference Method and RT-qPCR Methods

To compare the performance of the RT-qPCR method with the national reference method, 100 samples were simultaneously examined by both methods. Among them, 26 and 25 samples tested PRRSV positive while 74 and 75 samples were negative using the reference method and the developed method, respectively. Between the results of two methods, 24 positive and 73 negative samples were same. The sensitivity of the developed RT-qPCR method was 96.00% (24/25), and the specificity was 97.33% (73/75) with an overall coincidence rate of 97.00% (97/100). The kappa coefficient reached 0.97, suggesting a good consistency between the two methods (Table 5). Furthermore, there was no significant variation in the Ct values of the FAM and HEX channels between the national reference method and the RT-qPCR methods. The detailed data for Table 5 are shown in Appendix A.

### 3.11. Clinical Application

In total, 321 clinical samples were tested using the developed RT-qPCR method, which revealed that the positive detection ratio for the viral-solution, swabs (oral, nasal, and anal), tissues, environmental samples, and serum samples were 100% (20/20), 40% (24/60), 87% (91/105), 63% (60/96), and 88% (35/40), respectively (Figure 9a). To assess the reliability of the RT-qPCR method, we randomly selected 24.78% (57/230) of the positive samples for sequencing of the PRRSV ORF5 gene (Figure 9b). Basing on the RT-qPCR results, C-PRRSV2, HP-PRRSV2, and N-PRRSV2 were detected in 15.22% (35/230), 23.48% (54/230), and 61.30% (141/230) of the samples, respectively. To confirm the accuracy of RT-qPCR, sequencing was performed on a random subset of 57 samples that were positive for PRRSV by RT-qPCR. The sequencing results showed that the percentage of C-PRRSV2, HP-PRRSV2, and N-PRRSV2 in subtypes of PRRSV strains was 19.30% (11/57), 26.32% (15/57), and 54.39% (31/57), respectively, which was consistent with the RT-qPCR results. A clinical sample with dual infection of highly pathogenic PRRSV2 and NADC30-like strains was found using the RT-qPCR method and further confirmed by NSP2 sequencing, proving the RT-qPCR method could detect multiple strains in a sample.

Two of the clinical samples tested by the multiplex real-time PCR assay showed unexpected results, indicating the presence of mixed infections of different PRRSV strains. One sample (Sample 1) had Ct values of 27.42 in the FAM channel and 32.66 in the CY5 channel, corresponding to nucleic acid concentrations that differed by 7.99-fold from the standard curve. This suggested that the sample contained both N-PRRSV2 and C-PRRSV2 strains (Figure 10a). The another (Sample 2) sample had Ct values of 20.87 in the FAM channel and 30.45 in the HEX channel, corresponding to nucleic acid concentrations that differed by 15.76-fold from the standard curve. This suggested that the sample contained both N-PRRSV1 and C-PRRSV2 strains (Figure 10b).

After PCR amplification of the NSP2 gene and gel electrophoresis, two bands were observed in sample 1 (1041 and 648 bp) and sample 2 (951 and 648 bp) (Figure 10c). The four bands were gel purified for NSP2 gene sequencing, and the sequence alignment showed that the homology of two sequences amplicons (bands) in sample 1 to N-PRRSV2 and C-PRRSV2 was 90.90% and 91.70%. Likewise, those of two sequences amplicons in sample 2 to C-PRRSV2 and HP-PRRSV2 were 91.50% and 99.00% (Figure 10d). These results indicated that the RT-qPCR method can also detect mixed infections.

## 4. Discussion

In recent years, PRRS has become a major challenge in pig farming due to the emergence of various genotypes and subtypes of PRRSV [8,19,20]. In addition, multiple strain infection further complicates the prevention and control of PRRS in pig herds [21]. Most of the prevalent PRRSV strains in China are genotype 2, including the three main subtypes C-PRRSV2, HP-PRRSV2, and N-PRRSV2; genotype 1 PRRSV infection is comparatively less [22]. Therefore, it is important to develop convenient and reliable detection methods for identifying the various PRRSV strains currently circulating in pig farms.

Various detection methods are available for the detection and diagnosis of PRRSV infection, such as PCR, virus isolation, immunohistochemistry, and antibody detection [23]. Among them, PCR is the most commonly used method for detecting PRRSV, followed by sequencing to analyze genome characteristics [24]. Although PCR offers obvious advantages in clinical diagnosis, singular PCR, which requires the use of specific primers, is expensive and time-consuming, in in addition, it may yield false-negative results due to the genomic variations. Multiple PCR enables the concurrent amplification of numerous targeted genes within a single reaction mixture, utilizing a few specific pairs of primers. This approach is cost-effective butimproves the diagnostic efficiency. PCR requires frequent handling of reaction tubes, which increases the risk of contamination and potential false positive in the results. Furthermore, agarose gel electrophoresis to visualize the PCR amplicon does not allow proper quantitative analysis. These limitations can be addressed by employing fluorescent quantitative PCR, which enables precise quantification of target DNA molecules without the need for post-PCR processing.

At present, fluorescent quantitative PCR is primarily used to differentiate various viruses causing reproductive disorders, such as distinguishing PRRSV from African swine fever virus, classical swine fever virus, porcine parvovirus, and pseudorabies virus [25,26,27]. Some detection methods can differentiate classical strain from highly pathogenic PRRSV strains [28], while other detection methods can distinguish between the two genotypes PRRSV1 and PRRSV2 [29,30]. In this study, we developed a multiple real-time PCR method for the detection and identification of the four subtypes of PRRSV strains that exist in China. Our method exhibited high specificity against PRRSV1, three subtypes of PRRSV2. Moreover, this method revealed no cross-reactivity with other tested viruses causing reproductive and respiratory stress in pigs.

To improve the method of sensitivity and universality, primers and probes with concatenated degenerate bases that enable the detection of isolates with point mutations were designed. The probes were synthesized using the TaqMan technique, and the fluorescent probes at both ends of the primers were modified to efficiently capture the fluorescent signals after specific amplification of the gene region [16]. 

Notably, the method differs significantly from the national reference approach in its designation, which is exclusively devised for PRRSV2 and employs universal primers specific to the incredibly conserved region of the ORF6 gene, and distinguishing primers basing on the NSP2 gene segment to discriminate between the C-PRRSV2 and HP-PRRSV2 strains. In this study, ORF7 was employed since it has been proven to be a highly conserved gene and has been employed more frequently over the years [27,29,30], and the NSP2 gene fragment was employed as a marker to identify different virus strains, the same as the national reference method (GBT 35912-2018) [18,22,31]. These two genes are commonly used to establish new detection methods. Unlike the national reference method, the multiplex real-time PCR method can detect and differentiate between PRRSV1 and PRRSV2 genotypes, as well as the C-PRRSV2, HP- PRRSV2, and N-PRRSV2 strains. This is achieved through the implementation of universal primers targeting the conserved ORF7 gene region and discriminative primers specific to the variable domains of the NSP2 gene. We compared the method with the national reference method by testing 100 clinical samples and found that the former has excellent sensitivity and specificity. The overall coincidence rate in PRRSV2 detection is 97.00% while it cannot provide the coincidence rate of PRRSV subtyping.

The sensitivity of our method is significantly higher than that of the national reference method [32]. It allows rapid identification of viral infections within 2 h and covers various types of clinical samples including oral fluids, nasal swabs, anal swabs, fecal, blood, tissue, and even environmental samples [33]. In this study, 321 samples suspected of PRRSV infection were tested and were subsequently confirmed by ORF5 sequencing. The results indicated that serum and lung tissues are the clinical samples, with a higher PRRSV detection rate compared to other sample. The positive rate of serum samples was the highest at 88%, followed by lung samples at 87%, and environmental and rectal swab samples at 63% and 40%. This result is consistent with the fact that PRRSV has a prolonged survival time in the blood and targets the lung [34]. Furthermore, in this study shows that the NADC30-like strain has become the dominant strain, accounting for 61.30%, followed by highly pathogenic strains (23.48%) and the classic strains (15.22%). This is consistent with many researches reporting a gradual increase in the proportion of NADC30-like strains [7,35,36].

Our assay allows simultaneous detection of four subtypes of PRRSV strains in China and has potential applicability as an alternative diagnostic tool. However, the method only allows rapid detection of the ORF7 and NSP2 gene fragments and does not enable the identification of other genes. To determine the composition of the viral genome, whole genome sequencing would be needed. Furthermore, the genetic diversity and high mutability of PRRSV result in the frequent emergence of new recombinant strains. The rapid evolutionary rate and intricate recombination patterns of RNA viruses are serious challenges in developing new detection methods [8]. Evaluation of the PRRSV detection method on a larger number and a wider range of clinical samples is necessary to demonstrate its optimal effectiveness.

## 5. Conclusions

In conclusion, this study developed a clinically applicable detection and typing method for PRRSV strains, which offers several advantages in the context of PRRSV diagnosis. In particular, it allows for rapid strain typing, exhibits high sensitivity and specificity, and enables simultaneous detection of multiple PRRSV subtypes in a single reaction. Additionally, the method is capable of detecting PRRSV coinfection events. Overall, it would be a valuable reference method for PRRSV detection and identification of PRRSV subtypes.

## 6. Patents

A patent application has been submitted for this research, with the following numbers: application (202211143428.2), online public examination (CN116144836A), and authorization (CN 116144836 B).

## Figures and Tables

**Figure 1 viruses-15-01946-f001:**
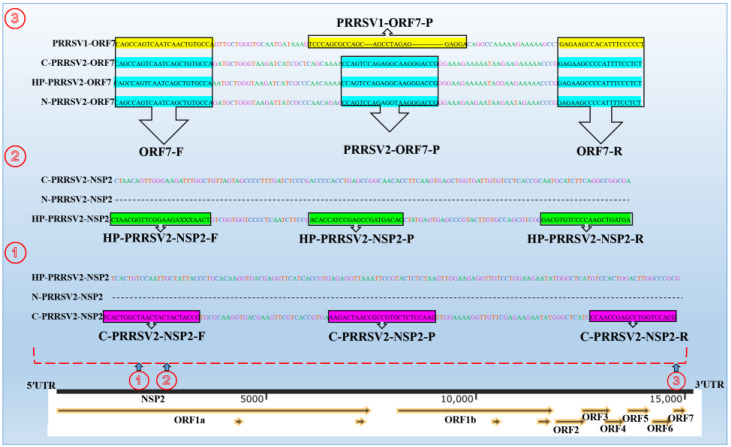
Primers and probes designation chart. Three gene fragments of PRRSV are named as ①, ②, and ③. Fragment 1 corresponds to the NSP2 region of classical strains, targeted by C-PRRSV2-NSP2-F/R/P primers and probes. Fragment 2 corresponds to another NSP2 region of highly pathogenic strains, targeted by HP-PRRSV2-F/R/P primers and probes. NADC30-like strains lack gene fragments in both NSP2 (① and ②) regions. Fragment 3, located in the ORF7 region, was targeted by PRRSV1-ORF7-P and PRRSV2-ORF7-P probes to distinguish between PRRSV1 and PRRSV2.

**Figure 2 viruses-15-01946-f002:**
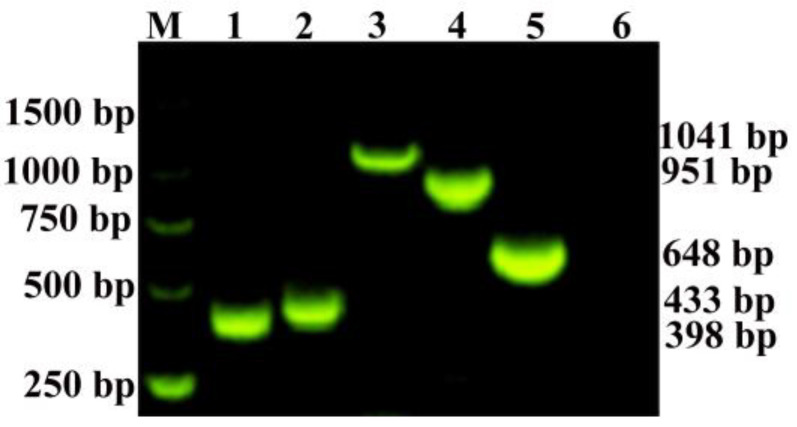
Results of conventional PCR detection of PRRSV ORF7 and NSP2 gene plasmids. M: DNA Marker (DL 2000 Marker), lanes 1–6 represent PRRSV1-ORF7, PRRSV2-ORF7, C-PRRSV2-NSP2, HP-PRRSV2-NSP2, N-PRRSV2-NSP2 plasmids, respectively, and the negative control.

**Figure 3 viruses-15-01946-f003:**
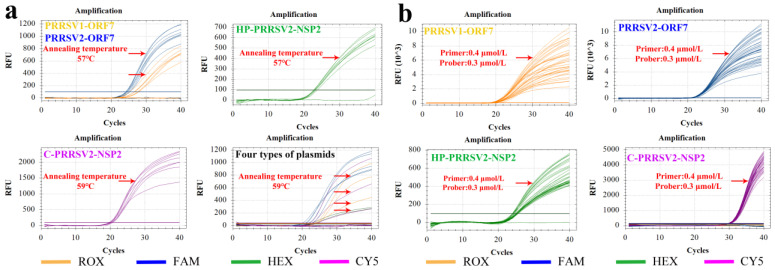
Optimization of annealing temperature and primer/probe concentrations in quadruple fluorescent RT–qPCR test. The optimal annealing temperature for the four virus plasmids was determined to be 59 °C. (**a**) Cross-validation of primers and probes for the four recombinant plasmids performed with varying concentrations. (**b**) Amplification results obtained from a concentration crossover experiment of primers and probes for the four recombinant plasmids. The optimal concentrations for the best amplification were 0.4 and 0.3 μmol/L of the primers and probes, respectively. The yellow, blue, green, and purple curves represent the amplification results of the PRRSV1-ORF7, PRRSV2-ORF7, HP-PRRSV2-NSP2, and C-PRRSV2-NSP2 plasmids, respectively. The red arrow indicates the curve of the optimal amplification.

**Figure 4 viruses-15-01946-f004:**
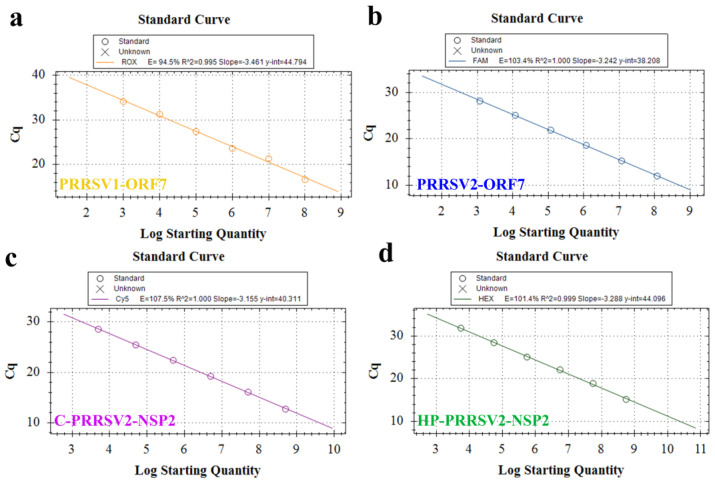
Establishment of the standard curves for recombinant plasmids from targeting virus strains. Standard curves are for the (**a**) PRRSV2-ORF7, (**b**) PRRSV1-ORF7, (**c**) C-PRRSV2-NSP2, and (**d**) HP-PRRSV2-NSP2 plasmids.

**Figure 5 viruses-15-01946-f005:**
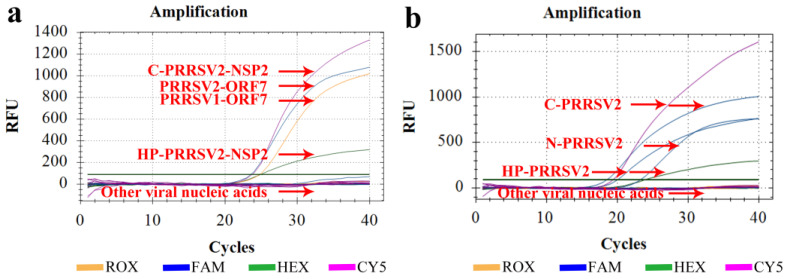
Specificity test results against various porcine viruses. (**a**) Nucleic acid extracts from four plasmids (PRRSV1-ORF7, PRRSV2-ORF7, C-PRRSV2-NSP2, and HP-PRRSV2-NSP2) and six viruses (CSFV, PEDV, PRV, PPV, PCV2, and PCV3) tested to validate the specificity of the RT–qPCR method. (**b**) Nucleic acid extracts from C-PRRSV2 (VR2332 strain), HP-PRRSV2 (TJ strain), N-PRRSV2 (NADC30-like strains), and six viruses—namely CSFV, PEDV, PRV, PPV, PCV2, and PCV3—tested to validate the specificity of the RT-qPCR method.

**Figure 6 viruses-15-01946-f006:**
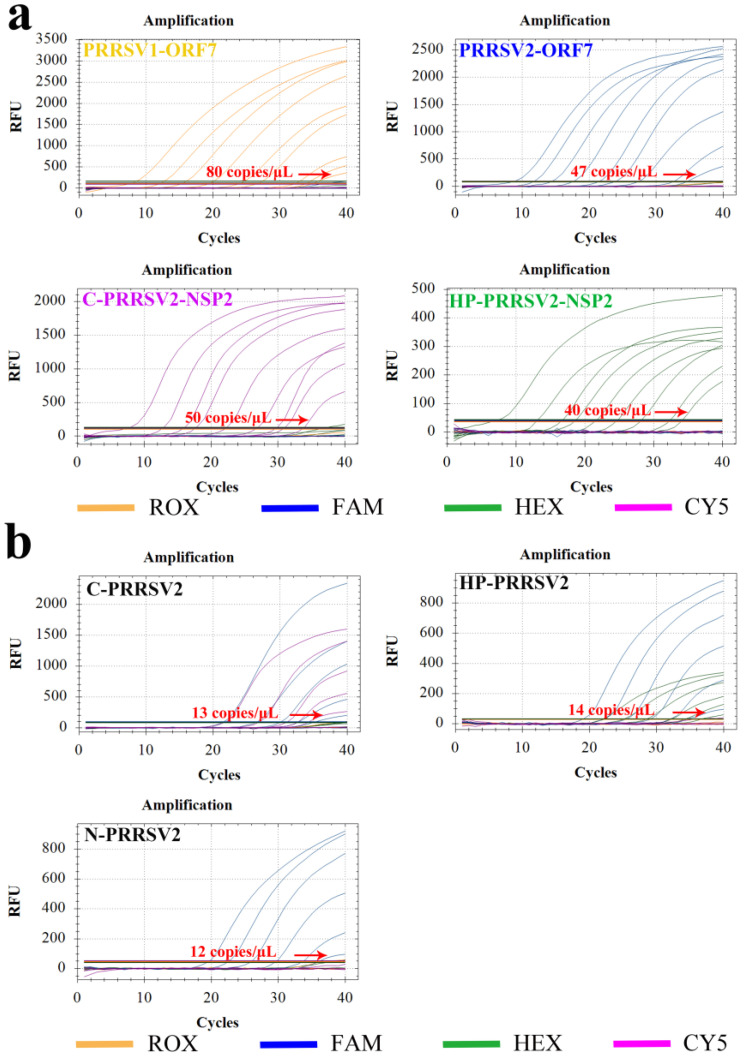
Minimal detection limits of the quadruple fluorescent quantitative PCR method. (**a**) Detection limits of PRRSV1-ORF7, PRRSV2-ORF7, HP-PRRSV2-NSP2, and C-PRRSV2-NSP2 plasmids: 80, 47, 50, and 40 copies/μL, respectively. (**b**) Minimum detection limits of the copy numbers for the three strains C-PRRSV2, HP-PRRSV2, and N-PRRSV2: 13, 14, and 12 copies/μL, respectively.

**Figure 7 viruses-15-01946-f007:**
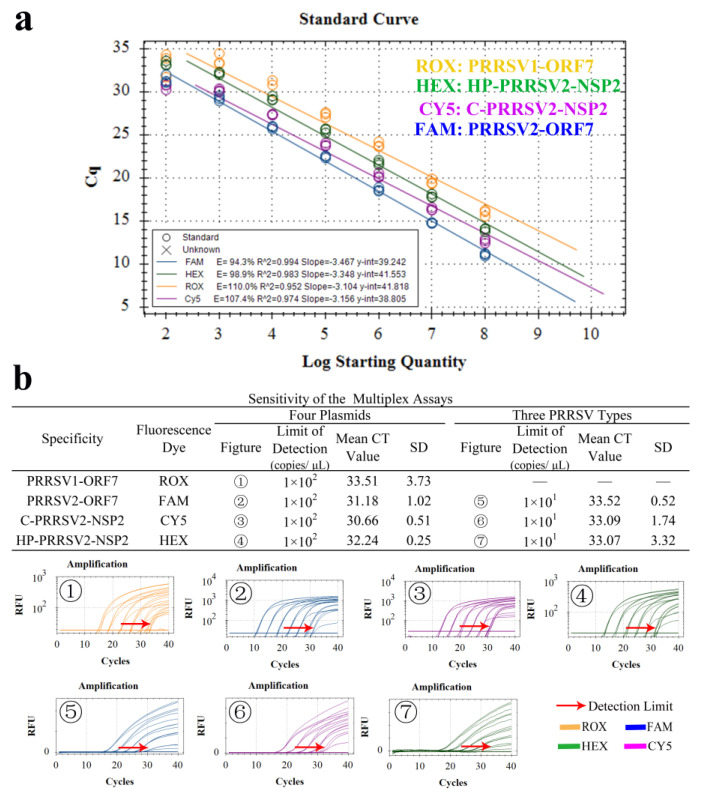
Standard curves and LODs. (**a**) Standard curves of the four plasmids. (**b**) Minimum limits of nucleic acid in detecting four plasmids (PRRSV1-ORF7, PRRSV2-ORF7, C-PRRSV2-NSP2, and HP-PRRSV2-NSP2) and three PRRSV (C-PRRSV2, HP-PRRSV2, and N-PRRSV2) subtypes.

**Figure 8 viruses-15-01946-f008:**
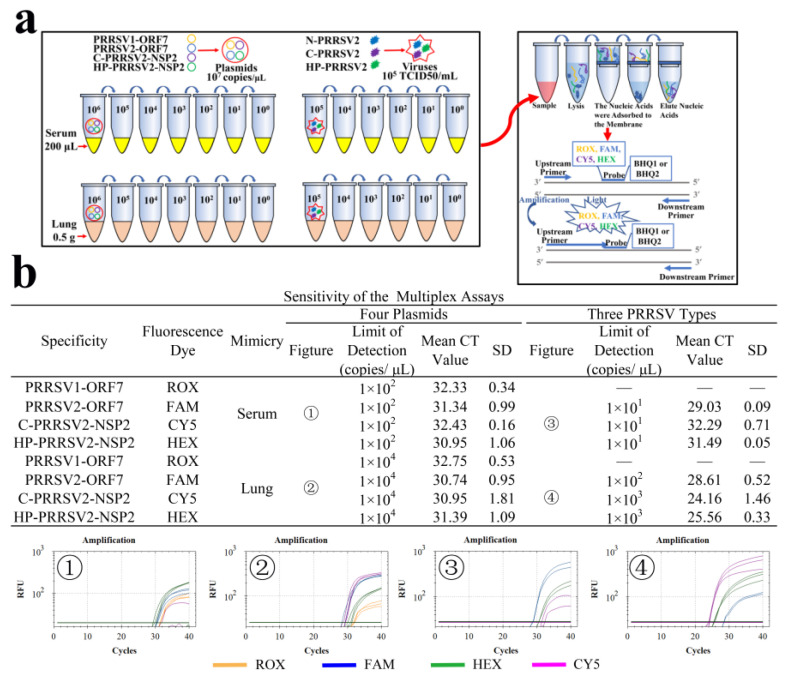
LODs of clinical simulants. (**a**) Processing steps for simulated clinical samples. (**b**) The LODs of serum samples and lung tissue samples simulated by plasmids or viruses.

**Figure 9 viruses-15-01946-f009:**
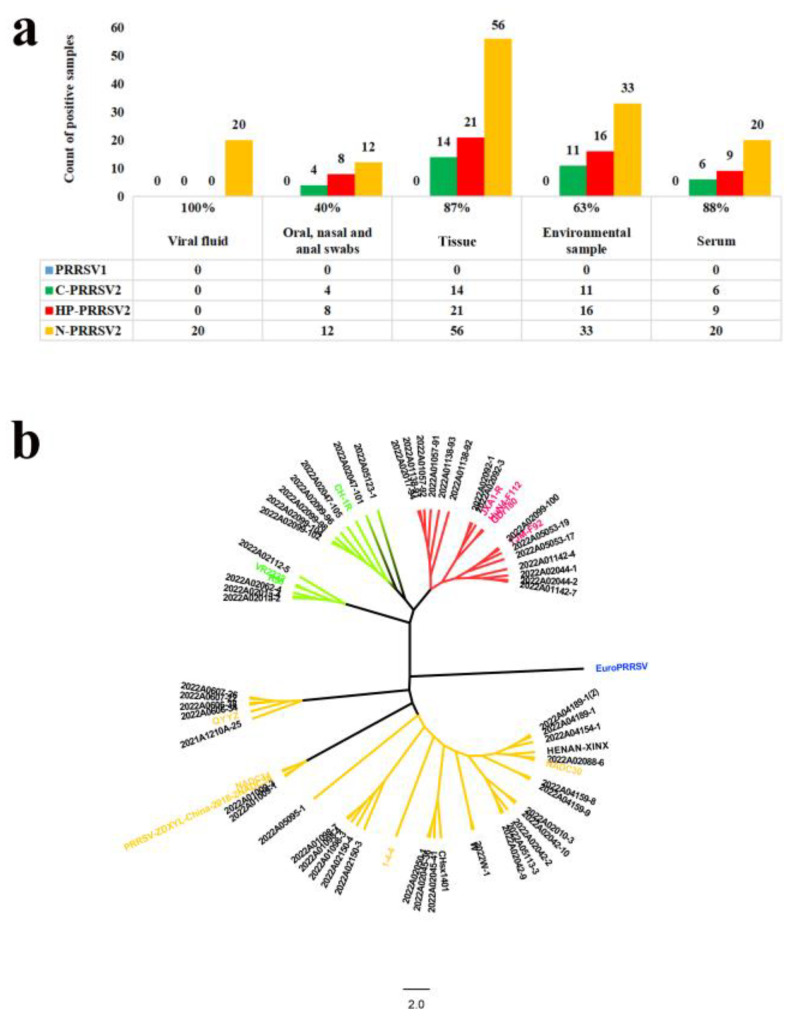
Clinical sample detection and PRRSV ORF5 sequencing results. (**a**) Genotyping and (**b**) sequencing results of five types of clinical samples are shown. Red, green, yellow, and blue branches represent the highly pathogenic PRRSV2, classic PRRSV2, NADC30-like PRRSV2, and PRRSV1 strains, respectively. NCBI reference strain names are represented by color-coded IDs, while sample names are marked in black IDs.

**Figure 10 viruses-15-01946-f010:**
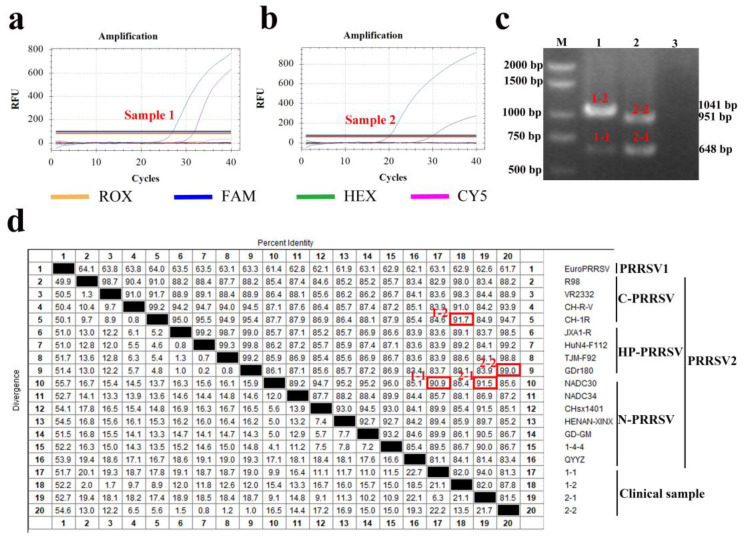
Clinical sample detection and identification results. (**a**) Detection results of sample 1: FAM and CY5 channel Ct values were 27.42 and 32.66, respectively. (**b**) Detection results of sample 2: FAM and HEX channel Ct values were 20.87 and 30.45, respectively. (**c**) NSP2 gene gel electrophoresis results for sample 1 and sample 2. M: DNA marker (DL 2000 Marker); Lane 1 and Lane 2 represent sample 1 and sample 2, and Lane 3 is the negative control. (**d**) Homology analysis of the NSP2 gene in sample 1 and sample 2 with reference strains: 1-1 and 1-2, representing larger and smaller size amplicons from sample 1, belonged to NADC30 and C-PRRSV2 strain, respectively; 2-1 and 2-2, representing larger and smaller size amplicons from sample 2, belong to NADC30 and HP-PRRSV2 strains, respectively. Within the target region, the red box represents the best sequence match with the highest sequence similarity.

**Table 1 viruses-15-01946-t001:** Plasmid amplification primers.

Genes	Primers	Sequences (5′-3′)	Amplicons Size (bp)	Upstream Primer Positions	Downstream Primer Positions	Reference
PRRSV (ORF7)	Forward Reverse	ATGGCCAGCCAGTCAATCA TCGCCCTAATTGAATAGGTG	398 (PRRSV1)	14,653–14,761	15,031–15,050	GB/T 1809-2023 [17]
			433 (PRRSV2)	14,933–14,951	15,346–15,365	
PRRSV (NSP2)	Forward Reverse	SGACACCTYCTTTGATTGG CTTGACARGGAGCTGCTTGA	1041 (C-PRRSV2)	2014–2032	3035–3051	Designed by our laboratory
			951 (HP-PRRSV2)	2184–2202	3115–3134	
			648 (N-PRRSV2)	2186–2240	2814–2833	

Note: The shown positions correspond to the genomes of the PRRSV1 LV isolate (GenBank ac-cession No. M96262), PRRSV2 CH-1a isolate (AY032626), classic PRRSV2 VR2332 isolate (EF536003), highly pathogenic PRRSV2 TJ isolate (EU860248), and PRRSV2 NADC30 isolate (JN654459).

**Table 2 viruses-15-01946-t002:** Primers and probes used in the developed RT-qPCR method.

Genes	Primers/Probes	Sequences (5′-3′)	Positions	Amplicon Size
PRRSV1 (ORF7)	PRRSV1-F	CAGCCAGTCAATCARCTGTGCCA	14,658–14,680	113 bp
	PRRSV1-R	AGRGGRAAATGKGGCTTCTC	14,751–14,770	
	PRRSV1-P	ROX-TCCCAGCGCCAGCARCCTAGRGGA-BHQ2	14,703–14,726	
PRRSV2 (ORF7)	PRRSV2-F	CAGCCAGTCAATCARCTGTGCCA	14,847–14,869	122 bp
	PRRSV2-R	AGRGGRAAATGKGGCTTCTC	14,949–14,968	
	PRRSV2-P	FAM-CCAGTCCAGAGGCAAGGGACCG-BHQ1	14,900–14,921	
C-PRRSV2 (NSP2)	C-PRRSV2-F	TCACTGGCTAACTACTACTACCG	2165–2187	134 bp
	C-PRRSV2-R	CGTGGACCAGGCTCGGTTGG	2279–2298	
	C-PRRSV2-P	CY5-AAGACTAACCGCCGTGCTCTCCAAG-BHQ2	2218–2242	
HP-PRRSV2 (NSP2)	HP-PRRSV2-F	CTAACGGTTCGGAAGAAACT	2762–2781	120 bp
	HP-PRRSV2-R	TCATCAGCTTGGGGACACGTC	2861–2881	
	HP-PRRSV2-P	HEX-GTGTCATCGGCTCGGATGGTGT-BHQ1	2806–2827	

Note: The shown positions correspond to the genomes of the PRRSV1 LV isolate (GenBank accession No. M96262), classic PRRSV2 VR2332 isolate (EF536003), highly pathogenic PRRSV2 TJ isolate (EU860248), and PRRSV2 NADC30 isolate (JN654459).

**Table 3 viruses-15-01946-t003:** Multiplex fluorescent quantitative PCR amplification reaction system.

Composition	Volume (μL)
Animal Detection U + Probe Master Mix	10.0
PRRSV1-ORF7-F	0.8
PRRSV1-ORF7-R	0.8
PRRSV1-ORF7-P	0.6
PRRSV2-ORF7-F	0.8
PRRSV2-ORF7-R	0.8
PRRSV2-ORF7-P	0.6
C-PRRSV2-NSP2-F	0.8
C-PRRSV2-NSP2-R	0.8
C-PRRSV2-NSP2-P	0.6
HP-PRRSV2-NSP2-F	0.8
HP-PRRSV2-NSP2-R	0.8
HP-PRRSV2-NSP2-P	0.6
Template	2.0
DEPC H_2_O	up to 25 μL
Total volume	25.0

Note: The concentrations of both the primer and probe were 10 µmol/L. RT-qPCR was performed with the following conditions: 50 °C for 2 min, 95 °C for 30 s, followed by 40 cycles of 95 °C for 15 s and 59 °C for 30 s.

**Table 4 viruses-15-01946-t004:** Intra- and inter-reproducibility of quadruple RT-PCR assay.

Target	Standard Sample (Copies/μL)	Intrabatch Repeatability Test		Interbatch Repeatability Test	
		Intrarepeatability Ct $\overset{-}{x}$ ± *SD*	Coefficients of Variation CV × 100%	Intrarepeatability Ct $\overset{-}{x}$ ± *SD*	Coefficients of Variation CV × 100%
PRRSV1-ORF7	8.77 × 10^5^	27.35 ± 0.04	0.10	27.61 ± 0.48	1.23
	8.77 × 10^4^	30.80 ± 0.84	1.93	31.2 ± 0.04	0.09
	8.77 × 10^3^	33.82 ± 0.01	0.21	33.92 ± 2.30	4.79
PRRSV2-ORF7	4.79 × 10^5^	27.02 ± 0.42	1.10	27.54 ± 0.62	1.59
	4.79 × 10^4^	31.69 ± 0.56	1.25	32.03 ± 1.24	2.74
	4.79 × 10^3^	34.53 ± 1.22	2.50	35.63 ± 0.98	1.94
C-PRRSV2-NSP2	4.71 × 10^5^	24.50 ± 1.24	3.58	24.35 ± 0.46	1.34
	4.71 × 10^4^	27.27 ± 0.76	1.97	28.67 ± 1.64	4.04
	4.71 × 10^3^	30.85 ± 0.92	2.11	31.75 ± 0.88	1.96
HP-PRRSV2-NSP2	5.54 × 10^5^	29.77 ± 1.64	3.90	30.17 ± 0.84	1.97
	5.54 × 10^4^	32.34 ± 1.17	3.80	31.66 ± 2.10	4.69
	5.54 × 10^3^	34.25 ± 0.84	1.73	35.5 ± 1.82	3.63

**Table 5 viruses-15-01946-t005:** Consistency comparison between the national reference method and the RT-qPCR method.

Kappa Test		Developed RT-qPCR		Total
		+	−	
National reference method	+	24	2	26
	−	1	73	74
total		25	75	100

## Data Availability

The data presented in this study are available in Appendix A here.

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
