# Peer review of "Development and Implementation of a Quadruple RT-qPCR Method for the Identification of Porcine Reproductive and Respiratory Syndrome Virus Strains"

_viruses, 2023, doi:10.3390/v15091946_

Round 1

Reviewer 1 Report

The study described herein is focused on the development of a multiplex RT-qPCR method for detecting and typing multiple PRRSV strains in a single reaction. Moreover, this new diagnostic assay seems to exhibit high sensitivity and specificity in clinical samples. 

Minor comments: 

1. Introduction, second paragraph:

"The analysis of the genome nucleotide sequences of PRRSV strains isolated from Europe and North America indicated a difference of about 44% among them" - please include reference

2. The quality of all figures needs to be improved - all seem blurry.

3. Two different fonts are used throughout the manuscript - please make sure to use the same font.

4. Discussion, last paragraph:

"Our allows simultaneous detection of four prevalent PRRSV strains in China..." - update to "Our assay allows simultaneous detection of four prevalent PRRSV strains in China..."

The quality of the English language is good. The authors seem to have used MJEditor for linguistic assistance as acknowledged in the manuscript.

Reviewer 2 Report

The presented manuscript is devoted to the development and validation of the PCR method, which makes it possible to detect the genome of various PRRS variants. The work is interesting in connection with the wide distribution of this pathogen throughout the world. The test was validated on a sufficiently large number of samples, in addition, the results were confirmed by sequencing. The manuscript is written in good English, I did not notice any significant errors. At the same time there are a few small remarks:

        -   page 2, in a sentence “In 1991, Wensvoort identified…” better to change word “pathogen” to “pathogenic”;

-          some discrepancies in the degree of dilutions used to test repeatability. On page 4 (2.6) indicated 10 in 4, 10 in 3, and 10 in 2 copies/μL, and in results (page 10) there is from 10 in 5 to 10 in 3 copies/μL;

-          page 14. In a sentence “Our allows simultaneous detection…” some word is missing. Maybe, “our test”” or “our method”;

-          page 9. Sensitivity test shows higher sensitivity with extracted RNA, than with plasmid dilutions. It is strange, because usually plasmids works better, due to high purity. Maybe, authors should test also 10 in 0 dilution after of 10 in 1, perhaps it can show even higher sensitivity.

Overall, manuscript contains interesting information, that can be used in practice. 

Reviewer 3 Report

The reviewed manuscript is dedicated to the design and validation of multiplex RT-qPCR assay detecting porcine reproductive and respiratory syndrome virus (PRRSV), a dangerous pathogen of pigs. The presented results are interesting for scientists, specializing on the field of molecular diagnostics. However, a number of issues needs to be addressed before publication.

Major issues:

1.      Authors are encouraged to add more information about PRRSV subtypes.

2.      Sections 2.2. Primers and probes 3.1. Design of primers and probes seem to be contradicting. In the first a specific primer pairs for NADC30-Like strains is mentioned, and in the second there are no such primers, as well in Figure 1 and Table 1. Thus, it is not clear in the presented manuscript, how was the third genotype detected.

3.      3.3. Preparation of standard templates — this section can be skipped in the Results or moved to Materials and Methods. Also, for serial dilutions, usage of carrier such as tRNA or linear polyacrylamide would prevent sorption of DNA on tube walls.

4.      Plausibly, a table with Cq values and annealing temperatures would be more readable than Figure 2a. The same goes for Figure 2b; actual Cq and RFU values for each combination would provide more information.

5.      3.5. Standard curve establishment — amplification efficiency for HP-PRRSV2-NSP2 and C-PRRSV2-NSP2 is higher than recommended by MIQE (10.1373/clinchem.2008.112797) 110% which compromises the quantification of cDNA. Probably, removing of 1-3 lowest concentration would make the efficacy acceptable.

Minor issues:

1.      Careful correction of the language is needed.

2.      2.1. Viruses and clinical samples — how were samples of other viral pathogens obtained?

3.      2.4. Establishment of the standard curve — it is necessary to add sequences of primers used to construct control plasmids

Figure 6 demonstrates phylogeny of the samples and their distribution in samples types, but not Cq values between two PCR methods. Possibly, one of figures is missing. Also, were the samples paired and taken from various tissues and organs of similar pigs or from different animals?

Minor language style corrections are needed.
